# Vision-and-Language Navigation Today and Tomorrow: A Survey in the Era of Foundation Models

**Yue Zhang**[1*]                                                   *zhan1624@msu.edu*

**Ziqiao Ma**[2*]                                                   *marstin@umich.edu*

**Jialu Li**[3*]                                                    *jialuli@cs.unc.edu*

**Yanyuan Qiao**[4*]                                     *yanyuan.qiao@adelaide.edu.au*

**Zun Wang**[3*]                                                    *zunwang@cs.unc.edu*

**Joyce Chai**[2†]                                                  *chaijy@umich.edu*

**Qi Wu**[4†]                                             *qi.wu01@adelaide.edu.au*

**Mohit Bansal**[3†]                                               *mbansal@cs.unc.edu*

**Parisa Kordjamshidi**[1†]                                        *kordjams@msu.edu*
[1] *Michigan State University*    [2] *University of Michigan*    [3] *UNC Chapel Hill*    [4] *University of Adelaide*
[*] *Equal Contribution* [†] *Equal Supervision*

**Reviewed on OpenReview:** *https://openreview.net/forum?id=yiqeh2ZYUh*

## Abstract

Vision-and-Language Navigation (VLN) has gained increasing attention over recent years and many approaches have emerged to advance their development. The remarkable achievements of foundation models have shaped the challenges and methods for VLN research. In this survey, we provide a top-down review that adopts a principled framework for embodied planning and reasoning, and emphasizes the current methods and future opportunities leveraging foundation models to address VLN challenges. We hope our in-depth discussions could provide valuable resources and insights: on the one hand, to document the progress and explore opportunities and potential roles for foundation models in this field, and on the other, to organize different challenges and solutions in VLN to foundation model researchers[1].

## 1 Introduction

Developing embodied agents that are capable of interacting with humans and their surrounding environments is one of the longstanding goals of Artificial Intelligence (AI) (Nguyen et al., 2021; Duan et al., 2022). These AI systems hold immense potential for real-world applications to serve as multi-functional assistants in daily life, such as household robots (Szot et al., 2021), self-driving cars (Hu et al., 2023), and personal assistants (Chu et al., 2023). One formal problem setting to advance this research direction is Vision-and-Language Navigation (VLN) (Anderson et al., 2018), a multimodal and cooperative task that requires the agent to follow human instructions, explore 3D environments, and engage in situated communications under various forms of ambiguity. Over the years, VLN has been explored in both photorealistic simulators (Chang et al., 2018; Savva et al., 2019; Xia et al., 2018) and real environments (Mirowski et al., 2018; Banerjee et al.,

---

[1]GitHub repository: `https://github.com/zhangyuejoslin/VLN-Survey-with-Foundation-Models`

2021), leading to a number of benchmarks (Anderson et al., 2018; Ku et al., 2020; Krantz et al., 2020) that each presents slightly different problem formulations.

Recently, *foundation models* (Bommasani et al., 2021), ranging from early pre-trained models like BERT (Kenton & Toutanova, 2019) to contemporary large language models (LLMs) and vision-language models (VLMs) (Achiam et al., 2023; Radford et al., 2021), have exhibited exceptional abilities in multimodal comprehension, reasoning, and cross-domain generalization. These models are pre-trained on massive data, such as text, images, audio, and video, and could further be adapted for a broad range of specific applications, including embodied AI tasks (Xu et al., 2024b). Integrating these foundation models into the VLN task marks a pivotal recent advancement for embodied AI research, demonstrated through significant performance improvements (Chen et al., 2021b; Wang et al., 2023h; Zhou et al., 2024a). Foundation models have also brought new opportunities to the VLN field, such as expanding the research focus from multi-modal attention learning and strategy policy learning to pre-training generic vision and language representations, hence enabling task planning, commonsense reasoning, as well as generalize to realistic environments.

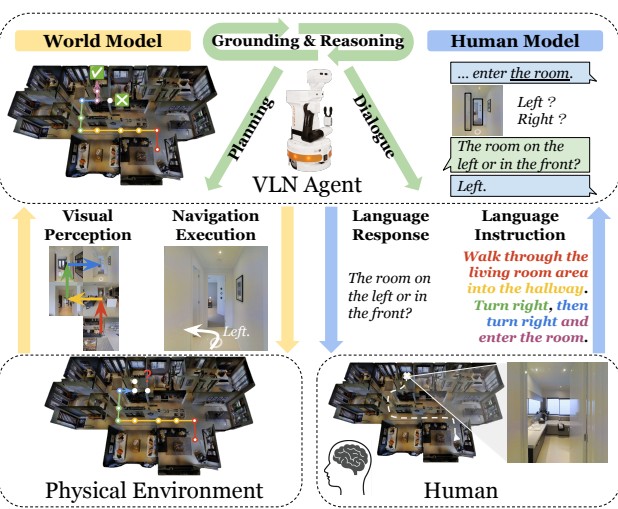

Figure 1: Organizing challenges and solutions in VLN using LAW framework (Hu & Shu, 2023).

Despite the recent impact of foundation models on VLN research, the previous surveys on VLN (Gu et al., 2022; Park & Kim, 2023; Wu et al., 2024) are from the pre-foundation-model era and mainly focus on the VLN benchmarks and conventional approaches, i.e., they are missing a comprehensive overview of the existing methods and opportunities leveraging foundation models to address VLN challenges. Especially with the emergence of LLMs, to the best of our knowledge, no review has yet discussed their applications in VLN tasks. Moreover, unlike previous efforts that discuss the VLN task as an isolated downstream task, the objective of this survey is twofold: **first**, to milestone the progress and explore opportunities and potential roles for foundation models in this field; **second**, to organize different challenges and solutions in VLN to foundation model researchers within a systematic framework. To build this connection, we adopt the LAW framework (Hu & Shu, 2023), where foundation models serve as backbones of *world model* and *agent model*. This framework offers a general landscape of reasoning and planning in foundation models, and is closely scoped with the core challenges in VLN.

Specifically, at each navigation step, the AI agents perceive the visual environment, receive language instructions from humans, and reason upon their representation of the world and humans to plan actions and efficiently complete navigation tasks. As shown in Figure 1, a ***world model*** is an abstraction that agents maintain to understand the external environment around them and how their actions change the world state (Ha & Schmidhuber, 2018; Koh et al., 2021). This model is part of a broader ***agent model***, which also incorporates a ***human model*** that interprets the instructions of its human partner, thereby informing the agent's goals (Andreas, 2022; Ma et al., 2023). To review the growing body of work in VLN and to understand the milestones achieved, we adopt a top-down approach to survey the field, focusing on fundamental challenges from three perspectives:

- Learning a world model to represent the visual environment and generalize to unseen ones.

- Learning a human model to effectively interpret human intentions from grounded instructions.

- Learning a VLN agent that leverages its world and human model to ground language, communicate, reason, and plan, enabling it to navigate environments as instructed.

| Name | World | | Human | | | VLN Agent | | | Dataset | |
|---|---|---|---|---|---|---|---|---|---|---|
| | Domain | Environment | Turn | Format | Gran. | Type | Act. Sp. | Other | Text | Route |
| LANI/CHAI (2018) | Indoors | CHALET | Single | Multi Instr | A | - | Disc | Mani | H | H |
| R2R (2018) | Indoors | Matterport3D | Single | Multi Instr | A | Robot | Graph | | H | P |
| R4R (2019) | Indoors | Matterport3D | Single | Multi Instr | A | Robot | Graph | | H | P |
| RxR (2020) | Indoors | Matterport3D | Single | Multi Instr | A | Robot | Graph | | H | P |
| SOON (2021a) | Indoors | Matterport3D | Single | Multi Instr | G | Robot | Graph | | H | P |
| REVERIE (2020b) | Indoors | Matterport3D | Single | Multi Instr | A, G | Robot | Graph | Detect | H | P |
| VNLA (2019) | Indoors | Matterport3D | Multi | Multi Instr | A, G | Robot | Graph | | T | P |
| HANNA (2019) | Indoors | Matterport3D | Multi | Multi Instr | A, G | Robot | Graph | | H | P |
| CVDN (2020) | Indoors | Matterport3D | Multi | Restricted | A | Robot | Graph | | H | H |
| VLN-CE (2020) | Indoors | Habitat, Matterport3D | Single | Multi Instr | A | Robot | Disc | | H | P |
| Robo-VLN (2021) | Indoors | Habitat, Matterport3D | Single | Multi Instr | A | Robot | Cont | | H | P |
| RobotSlang (2021) | Indoors | Real | Multi | Freeform | A | Robot | Disc | | H | P |
| ALFRED (2020) | Indoors | AI2-THOR | Single | Multi Instr. | A, G | Robot | Disc | Mani | H | P |
| TEACh (2022) | Indoors | AI2-THOR | Multi | Freeform | A, G | Robot | Disc | Mani | H | H |
| DialFRED (2022) | Indoors | AI2-THOR | Multi | Restricted | A, G | Robot | Disc | Mani | H, T | P |
| TouchDown (2019) | Outdoors | Google Street View | Single | Multi Instr | A | - | Graph | | H | P |
| Street Nav (2020) | Outdoors | Google Street View | Multi | Multi Instr | A | - | Disc | | T | P |
| Talk2Nav (2021) | Outdoors | Google Street View | Single | Multi Instr | A, G | - | Disc | | H | P |
| TtW (2018) | Outdoors | Real | Multi | Freeform | A, G | - | Disc | | H | H |
| LCSD (2019) | Outdoors | CARLA | Single | Multi Instr | A | Driving | Disc | | H | P |
| CDNLI (2020) | Outdoors | CARLA | Multi | Multi Instr | A, G | Driving | Cont | | H, T | H |
| SDN (2022) | Outdoors | CARLA | Multi | Freeform | A, G | Driving | Disc, Cont | | H | H |
| AerialVLN (2023b) | Outdoors | AirSim | Single | Multi Instr | A, G | Aerial | Disc | | H | H |
| ANDH (2023a) | Outdoors | xView | Multi | Freeform | A, G | Aerial | Disc | | H | H |

Table 1: A summary of existing VLN benchmarks, taxonomized based on several key aspects: the **world** in which navigation occurs, the type of **human** interaction involved, the action space and tasks assigned to the **VLN agent**, and the methods of **dataset** collection. For the **world**, we consider their **domain** (either indoors or outdoors) and the **environment**. For the **human**, we consider their **turns of interaction** (either single or multiple turn), the **format of communication** (freeform dialogue, restricted dialogue, or multiple instructions), and the **language granularity** (action-directed and goal-directed). For the **VLN agent**, we consider their **agent types** (e.g., household robot, autonomous driving vehicles, or autonomous aerial vehicles), their **action space** (graph-based, discrete or continuous), and **other additional tasks** (manipulation and object detection). For **dataset collection**, we consider the **text collection** (by human or templated) and the **route demonstrations** (by human or planner).

We present a hierarchical and fine-grained taxonomy in Figure 2.2 to discuss challenges, solutions, and future directions based on foundation models for each model. To organize this survey, we start with a brief overview of the background and related research efforts as well as the available benchmarks in this field (§2). We structure the review around how the proposed methods have addressed the three key challenges described above: world model (§3), human model (§4), and VLN agent (§5). Finally, we discuss the current challenges and future research opportunities, particularly in light of the rise of foundation models (§6).

## 2 Background and Task Formulations

In this section, we discuss the background, clarify the scope of this survey, define the VLN problem, and briefly overview the benchmarks.

### 2.1 Cognitive Underpinnings of VLN

Humans and other navigational animals demonstrate early understanding and strategies for navigating their environments (Rodrigo, 2002; Brand et al., 2015; Lingwood et al., 2018). For example, Gallistel (1990) describes two basic mechanisms: *piloting*, which involves environmental landmarks and computes distances and angles; and *path integration*, which calculates displacement and orientation changes through self-motion sensing. Central to understanding spatial navigation is the *cognitive map hypothesis*, suggesting that the brain forms a unified spatial representation to support memory and guide navigation (Epstein et al., 2017; Bellmund et al., 2018). For instance, Tolman (1948) observed that rats could adopt the correct novel path when familiar paths are blocked and landmarks are absent. Neuroscientists also discovered hippocampal

place cells, indicating a spatial coordinate system that encodes landmarks and goals allocentrically (O'Keefe & Dostrovsky, 1971; O'keefe & Nadel, 1978). Recent studies also propose non-Euclidean representations, *e.g.*, *cognitive graphs*, which illustrate the complexity of how we represent spatial knowledge of the world (Warren, 2019; Ericson & Warren, 2020). While visual and auditory perceptions are obviously integral to spatial representation (Klatzky et al., 2006), our linguistic skills and spatial cognition are also closely intertwined (Pruden et al., 2011). For instance, researchers have shown that understanding different aspects of spatial language can help with space-related tasks (Pyers et al., 2010), and that language influences how children interact with space by assisting them to recognize the importance of landmarks in identifying locations (Shusterman et al., 2011). Studying VLN not only enhances the development of embodied AI that follows human instructions in visual environments, but also deepens our understanding of how cognitive agents develop navigation skills, adapt to different environments, and how language use is connected to visual perceptions and actions.

## 2.2 Relevant Tasks and Scope of the Survey

Following natural language navigation instructions has traditionally been modeled using symbolic world representations such as maps (Anderson et al., 1991; MacMahon et al., 2006; Paz-Argaman & Tsarfaty, 2019). However, our survey focuses on models that employ visual environments and address the challenges of multimodal understanding and grounding. Likewise, we redirect readers to extensive surveys on visual navigation (Zhu et al., 2021b; Zhang et al., 2022a; Zhu et al., 2022) and mobile robot navigation (Gul et al., 2019; Crespo et al., 2020; Möller et al., 2021), which concentrate on visual perception and physical embodiment. However, these studies provide minimal discussions on the role of language in navigation tasks. While we inevitably extend our discussions of VLN to encompass areas beyond navigation, such as mobile manipulation and dialogue, our primary focus remains on navigational tasks, for which we provide a detailed literature review. Besides, unlike previous VLN surveys (Gu et al., 2022; Park & Kim, 2023; Wu et al., 2024), which offer a bottom-up summary focusing on benchmarks and modeling innovations, our survey adopts a top-down approach, and uses the roles of foundation models to categorize the research efforts into three fundamental challenges from the aspects of the world model, the human model, and the VLN agent. Note that this survey concentrates on frontier methods associated with the rise of foundation models. Thus, we point to the earlier generation of models (*e.g.*, LSTM-based methods) very briefly at the beginning of each section to motivate our discussions.

## 2.3 VLN Task Formulations and Benchmarks

**VLN Task Definition.** A typical VLN agent receives a (sequence of) language instruction(s) from human instructors at a designated position. The agent navigates through the environment using an egocentric visual perspective. By following the instructions, its task is to generate a trajectory over a sequence of discrete views or lower-level actions and control (*e.g.*, FORWARD 0.25 meter) to reach the destination, which is considered successful if the agent arrives within a specified distance (*e.g.*, 3 meters) from the destination. Besides, the agent may exchange information with the instructor during navigation, either by requesting help or engaging in freeform language communication. Additionally, there has been an increasing expectation for VLN agents to integrate additional tasks such as manipulation (Shridhar et al., 2020) and object detection (Qi et al., 2020b), along with navigation.

**Benchmarks.** Unlike other multimodal tasks such as VQA, which have a relatively fixed task definition and format, VLN encompasses a wide range of benchmarks and task formulations. These distinctions introduce unique challenges in addressing the broader VLN task and must be clearly understood as prerequisites for developing effective methods with appropriate foundation models. As is summarized in Table 1, existing VLN benchmarks can be taxonomized based on several key aspects in the LAW framework: (1) the **world** where navigation occurs, including the domain (indoors or outdoors) and the specifics of the environment. (2) the type of **human** interaction involved, including the interaction turns (single or multiple), communication format (freeform dialogue, restricted dialogue, or multiple instructions), and language granularity (action-directed or goal-directed). (3) the **VLN agent**, including its types (*e.g.*, household robots, autonomous driving vehicles, or autonomous aerial vehicles), action space (graph-based, discrete, or continuous), and additional tasks (manipulation and object detection). (4) the **dataset** collection, including text

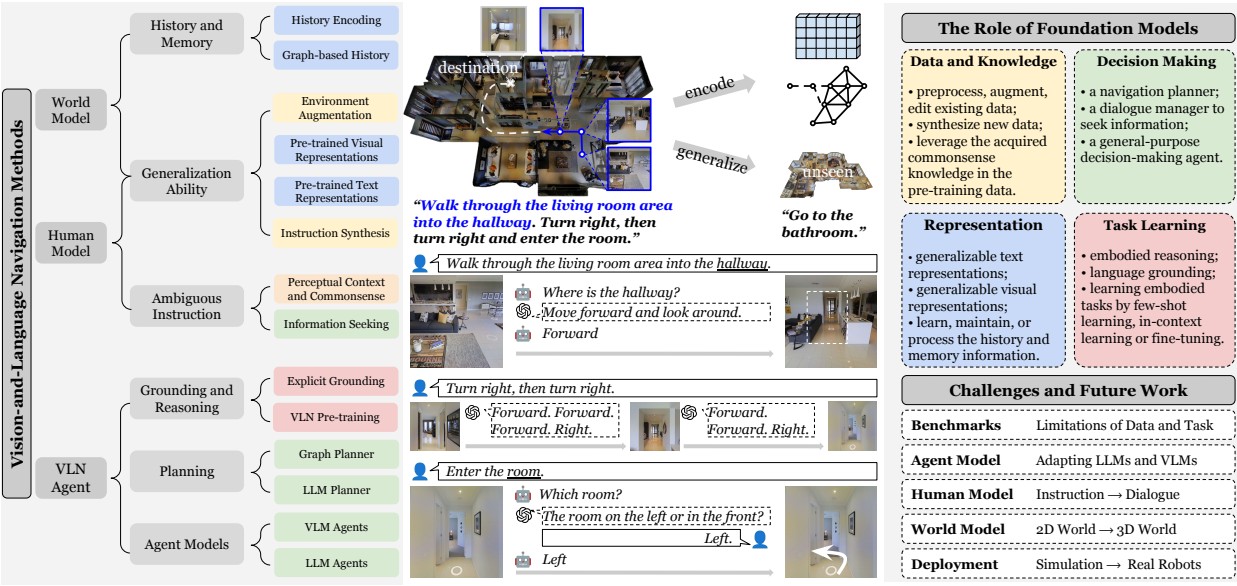

Figure 2: VLN challenges and solutions within the framework of world model, human model, and VLN agent. We discuss history and memory in the world model, ambiguous instructions in the human model, generalization ability in them both. For the VLN agent, we discuss methods for grounding and reasoning, planning, and adapting foundation models as agents. Depending on the role served by the foundation models, we categorize these methods into four types. Additionally, we discuss the potential future of the foundation model for the VLN task.

collection method (human-generated or templated) and route demonstrations (human-performed or planner-generated). Representatively, Anderson et al. (2018) create the Room-to-Room (R2R) dataset based on the Matterport3D simulator (Chang et al., 2018), where an agent needs to follow fine-grained navigation instructions to reach the goal. Room-across-Room (RxR) (Ku et al., 2020) is a multilingual variation, including English, Hindi, and Telugu instructions. It offers a larger sample size and provides time-aligned instructions for virtual poses, enriching the task's linguistic and spatial information. Matterport3D allows VLN agents to operate in a *discrete environment* and rely on pre-defined connectivity graphs for navigation, where agents travel on the graph by teleportation between adjacent nodes, referred to as VLN-DE. To make the simplified setting more realistic, Krantz et al. (2020); Li et al. (2022c); Irshad et al. (2021) propose VLN in *continuous environments* (VLN-CE) by transferring discrete R2R paths to continuous spaces (Savva et al., 2019). Robo-VLN (Irshad et al., 2021) further narrows the sim-to-real gap by introducing VLN with continuous action spaces that are more realistic in robotics settings. Recent VLN benchmarks have undergone several design changes and expectations, which we discuss in § 6.

**Evaluation Metrics.** Three main metrics have been employed to evaluate navigation wayfinding performance (Anderson et al., 2018): (1) *Navigation Error (NE)*, the mean of the shortest path distance between the agent's final position and the goal destination; (2) *Success Rate (SR)*, the percentage of the final position being close enough to the goal destination; and (3) *Success Rate Weighted Path Length (SPL)*, which normalizes success rate by trajectory length to balance both the success rate in reaching the correct destination and the efficiency of the path. Some other metrics are used to measure the faithfulness of instruction following and the fidelity between the predicted and the ground-truth trajectory, for example: (4) *Coverage Weighted by Length Score (CLS)* (Jain et al., 2019) measures how closely an agent's trajectory follows the reference path. It balances two key aspects of the agent's performance: the extent of coverage of the reference path and the efficiency of the agent's navigation by considering the length score; (5) *Normalized Dynamic Time Warping (nDTW)* (Ilharco et al., 2019), which penalizes deviations from the ground-truth trajectories; and (6) *Normalized Dynamic Time Warping Weighted by Success Rate (sDTW)* (Ilharco et al., 2019), which penalizes deviations from the ground-truth trajectories and also considers the success rate.

## 2.4 Foundation Models

Foundation models are trained on large-scale datasets, which show strong generalization capability for a wide range of downstream applications. Text-only foundation models, such as pre-trained language models like BERT (Kenton & Toutanova, 2019) and GPT-3 (Brown et al., 2020), have revolutionized the field of NLP by setting new benchmarks for tasks like text generation, translation, and understanding. Building on the success of these models, vision-language (VL) foundation models, like LXMERT (Tan & Bansal, 2019), CLIP (Radford et al., 2021) and GPT-4 (Achiam et al., 2023), have expanded the paradigm to multimodal learning by integrating both visual and textual data, proving particularly impactful in various VL applications (Li et al., 2019a; Ramesh et al., 2021; Alayrac et al., 2022; Hong et al., 2021; Zhang et al., 2025; Cheng et al., 2024; Kamali & Kordjamshidi, 2023). For a more comprehensive overview of foundation models and their applications, we encourage readers to refer to existing survey papers such as Bommasani et al. (2021), Du et al. (2022), and Zhou et al. (2023).

# 3 World Model: Learning and Representing the Visual Environments

A world model helps the VLN agent to understand their surrounding environments, predict how their actions would change the world state, and align their perception and actions with language instructions. Two challenges have been highlighted in existing work about learning a world model: encoding the visual history of observations within the current episode as memory, and achieving generalization to unseen environments.

## 3.1 History and Memory

Different from other vision-language tasks like Visual Question Answering (VQA) (Antol et al., 2015), Visual Entailment (Xie et al., 2019), the VLN agent needs to incorporate the history information of past actions and observations into its current step's input to determine the action rather than solely considering the image and text in a single step. Prior to employing the foundation models in VLN, LSTM hidden states served as an implicit memory supporting agents' decision-making during navigation, and researchers further design different attention mechanisms (Tan et al., 2019; Wang et al., 2019) or auxiliary tasks (Ma et al., 2019; Zhu et al., 2020) to improve the alignment between the encoded history and instructions.

**History Encoding.** Different techniques have been proposed to encode navigation history using foundation models. A multi-modal Transformer is built upon encoded instructions and navigation history for decision-making, which is usually initialized from a model pre-trained on in-domain instruction-trajectory data like Prevalent (Hao et al., 2020). Some approaches encode the navigation history in recurrently updated state tokens. Hong et al. (2021) proposes to utilize a single `[CLS]` token from last step for encoding the history information, while Lin et al. (2022a) introduces a variable-length memory framework that stores multiple action activations from previous steps in a memory bank as the history encoding. Despite their effectiveness, these methods are limited by the need for step-by-step token updates, making it challenging to efficiently retrieve history encodings at arbitrary steps in the navigation trajectory, which can hinder scalability in pre-training. Another line of work directly encodes navigation history as a sequence with multi-modal Transformer. Among them, Pashevich et al. (2021) encodes single-view images for each step in a trajectory. Chen et al. (2021b) further proposes a panorama encoder to encode the panoramic visual observation at each time step, followed by a history encoder to encode all the past observations. This hierarchical design separately processes the spatial relationship in a panoramic view and the temporal dynamics across panoramas in the navigation history. Besides, this method eliminates the dependency on recurrently updated state tokens for history encoding, facilitating efficient and large-scale pre-training on instruction-path pairs. Follow-up research replaces the panorama encoder with mean pooling of images (Kamath et al., 2023) or front-view image encoding (Qiao et al., 2022), both maintaining effective navigation performance. With the advent of LLM-based navigation agents, some works (Zhou et al., 2024b) focus on converting the visual environment into textual descriptions, and explaining the world with text became the trend. The navigation history is then encoded as a sequence of these image descriptions, along with relative spatial information such as heading, elevation, and distance. HELPER (Sarch et al., 2023) designs an external

memory of language-program pairs that parses the free-form human-robot dialogue into action programs through retrieval-augmented LLM prompting.

**Graph-based History.** Another line of research enhances the navigation history modeling with graph information. For example, some of these techniques utilize a structured Transformer encoder to capture the geometric cues in the environment (Chen et al., 2022c; Deng et al., 2020; Wang et al., 2023b; Zhou & Mu, 2023; Su et al., 2023; Zheng et al., 2024b; Wang et al., 2021; Chen et al., 2021a; Zhu et al., 2021a). In addition to the topological graph used in encoding, many works propose to include the top-down view information (*e.g.*, grid map (Wang et al., 2023g; Liu et al., 2023a), semantic map (Hong et al., 2023a; Huang et al., 2023a; Georgakis et al., 2022; Anderson et al., 2019; Chen et al., 2022a; Irshad et al., 2022), local metrics map (An et al., 2023)), and local neighborhood map (Gopinathan et al., 2023) in modeling the observation history during navigation. Recent advances in LLM-based navigation agents have introduced innovative approaches to memory construction using maps. For instance, Chen et al. (2024a) proposes a novel map-guided GPT-based agent that utilizes a linguistical-formed map to store and manage topological graph information. MC-GPT (Zhan et al., 2024b) introduces a topological map as the memory structure to record information about viewpoints, objects, and their spatial relationships.

## 3.2 Generalization across Environments

One main challenge in the VLN is learning from limited available environments and generalizing to new and unseen environments. Many works demonstrate that learning from semantic segmentation features (Zhang et al., 2021a), dropout information in the environment during training (Tan et al., 2019), and maximizing the similarity between semantically-aligned image pairs from different environments (Li et al., 2022a) improve agents' generalization performance to unseen environments. These observations suggest the need to learn from large-scale environment data to avoid overfitting to training environments. Next, we discuss how existing works collect new environment data, and utilize it in training.

**Pre-trained Visual Representations.** Most works obtain vision representations from ResNet pre-trained on ImageNet (Anderson et al., 2018; Tan et al., 2019). Shen et al. (2022) replace ResNet with the CLIP visual encoder (Radford et al., 2021), which is pre-trained with contrastive loss between image-text pairs and naturally better aligns the image with the instructions, boosting the VLN performance. Wang et al. (2022b) further explores transferring vision representation learned from video data for VLN task, suggesting that temporal information learned from video is crucial for navigation.

**Environment Augmentation.** One main line of research focuses on augmenting the navigation environment with auto-generated synthetic data. EnvEdit (Li et al., 2022b), EnvMix (Liu et al., 2021), KED (Zhu et al., 2023), and FDA (He et al., 2024a) generate synthetic data by changing the existing environments from Matterport3D. Specifically, they mix up rooms from different environments, change the appearance and style of the environments, and interpolate high-frequency features with the environments. Pathdreamer (Koh et al., 2021) and SE3DS (Koh et al., 2023) further synthesize the environments in future steps given current observations and explore utilizing the synthesis view as augmented data for VLN training.

The learning paradigm from the collected environments has changed with the advances in foundation models. Prior to the prevalence of pre-training in foundation models, most works directly augment the training environment with the auto-collected new environments and fine-tune a LSTM-based VLN agent (Li et al., 2022b; Liu et al., 2021; Koh et al., 2021; 2023; Zhu et al., 2023). As pre-training has been demonstrated to be crucial for foundation models, it has also become a standard practice in VLN to learn from collected environments during the pre-training stage (Li & Bansal, 2024; Kamath et al., 2023; Chen et al., 2022b; Wang et al., 2023h; Lin et al., 2023b; Guhur et al., 2021a; He et al., 2024a). Large-scale pre-training with augmented in-domain data has become crucial in bridging the gap between agents' and humans' performance. The in-domain pre-trained multi-modal transformer has been proven to be more effective than the multi-modal Transformer initialized from VLMs, like Oscar (Li et al., 2020) and LXMERT.

# 4 Human Model: Interpreting and Communicating with Humans

In addition to learning and modeling the world, VLN agents need a human model that comprehends human-provided natural language instructions per situation to complete navigation tasks. There are two main challenges: resolving ambiguity and generalization of grounded instructions in different visual environments.

## 4.1 Ambiguous Instructions

Ambiguous instructions mainly arise in single-turn navigation scenarios, where the agent follows an initial instruction without further human interaction for clarification. These instructions lack the flexibility to train the agent to adapt its language understanding and visual perception to the dynamic environments. For instance, instructions may contain landmarks invisible at the current view or indistinguishable landmarks visible from multiple views (Zhang & Kordjamshidi, 2023). The issue of ambiguous instructions is barely addressed before the application of foundational models to VLN. Although LEO (Xia et al., 2020) attempts to aggregate multiple instructions to describe the same trajectory from different perspectives, it still relies on human-annotated instructions. However, comprehensive perceptual context and commonsense knowledge from foundational models enable the agent to interpret ambiguous instructions using external knowledge, as well as seek assistance from other human models.

**Perceptual Context and Commonsense Knowledge.** Large-scale cross-modal pre-trained models like CLIP are capable of matching visual semantics with text. This enables the VLN agent to utilize information from the visual objects and their states in the current perception to resolve ambiguity, especially in single-turn navigation scenarios. For example, VLN-Trans (Zhang & Kordjamshidi, 2023) constructs easy-to-follow sub-instructions with visible and distinctive objects obtained from CLIP to pre-train a Translator that converts original ambiguous instructions into easily understandable sub-instruction representations. LANA+ (Wang et al., 2023f) leverages CLIP to query a text list of landmark semantic tags with the visual panoramic observations, and selects the top-ranked retrieved textual cues as representations of the salient landmarks to follow. KERM (Li et al., 2023a) proposes a knowledge-enhanced reasoning model to retrieve facts where knowledge is described by language descriptions for the navigation views. NavHint (Zhang et al., 2024b) constructs a hint dataset, providing detailed visual descriptions to help the VLN agent build a comprehensive understanding of the visual environment rather than focusing solely on the objects mentioned in the instructions. On the other hand, the commonsense reasoning ability of LLMs can be used to clarify or correct ambiguous landmarks in the instructions, and break instructions into actionable items. For example, Lin et al. (2024b) use LLMs to provide commonsense about open-world landmark co-occurrences and conduct CLIP-driven landmark discovery accordingly. SayCan (Ahn et al., 2022) breaks an instruction into a ranked list of pre-defined admissible actions and combines them with an affordance function that assigns higher weights to the objects appearing in the current scene.

**Information Seeking.** While ambiguous instructions can be resolved based on visual perception and situational context, another more direct approach is to seek help from the communication partner, i.e., the human speakers who generate the instructions (Nguyen & Daumé III, 2019; Paul et al., 2022). There are three key challenges in this line of work: (1) deciding when to ask for help (Chi et al., 2020); (2) generating information-seeking questions, *e.g.*, next action, objects, and directions (Roman et al., 2020; Singh et al., 2022); (3) developing an oracle that provides the queried information, which could be either real humans (Singh et al., 2022), rules and templates (Gao et al., 2022), or neural models (Nguyen & Daumé III, 2019). LLMs and VLMs could potentially fit two roles in this framework, either as information-seeking models, or as proxies for human helpers or information-providing models. Preliminary research has explored the use of LLMs as the information-seeking model, addressing determining both when and what to ask. This is achieved with the help of techniques including conformal prediction (CP) (Ren et al., 2023) or in-context learning (ICL) (Chen et al., 2023c). For the latter, foundation models play the role of a helper who has access to oracle information, such as the location of the destination and a map of the environment, which is not available to the task performer. Very recently, VLN-Copilot (Qiao et al., 2024) enables agents to actively seek assistance when encountering confusion, with the LLM serving as a copilot to facilitate navigation. Fan et al. (2023b) demonstrate that GPT-3 can decompose ground-truth responses in the training data

step-by-step, which helps in training an oracle model using a pre-trained SwinBert (Lin et al., 2022b) video-language model. They also demonstrate large vision-language models like mPLUG-Owl (Ye et al., 2023) can serve as strong zero-shot oracles off the shelf. In addition, self-motivated communication agents have been developed (Zhu et al., 2021c) by learning the confidence of the oracle to produce a positive answer, which enables a self-Q&A manner where the oracle can be removed at inference time.

## 4.2 Generalization of Grounded Instructions

The limited scale and diversity of navigation data is another significant issue affecting the VLN agent's ability to comprehend various linguistic expressions and follow instructions effectively, particularly in unseen navigation environments. Although the language style itself has good generalization capability across seen and unseen environments (Zhang et al., 2021a), how to ground the instructions with the unseen environments is potentially a hard task given the limited scale of training instructions. Foundation models help address these issues through both pre-trained representations and instruction generation for data augmentation.

**Pre-trained Text Representations.** Before the foundation models, many works rely on text encoders, such as LSTM, to represent text instructions (Anderson et al., 2018; Tan et al., 2019). The foundation models significantly enhance the VLN agent's language generalization ability through pre-trained representations. For example, PRESS (Li et al., 2019b) fine-tunes the pre-trained language model BERT (Kenton & Toutanova, 2019) to obtain text representations that generalize better to previously unseen instructions. The multi-modal Transformers (Tan & Bansal, 2019; Lu et al., 2019) boost methods, such as VLN-BERT (Majumdar et al., 2020) and PREVALENT (Hao et al., 2020), to obtain more generic vision-linguistic representations by pre-training on large-scale text-image pairs collected from the web. Airbert (Guhur et al., 2021b) trains ViLBERT-like architecture to learn text representations from image-caption pairs collected from the Internet. CLEAR (Li et al., 2022a) learns cross-lingual language representations that capture the visual concepts behind the instruction. ProbES (Liang et al., 2022) self-explores environments by sampling trajectories and automatically constructs the corresponding instruction by filling the instruction templates with movements and object phrases detected by CLIP. Additionally, it leverages prompt-based learning to facilitate fast adaptation of language embeddings. NavGPT-2 (Zhou et al., 2025) explores leveraging vision-and-language representations from pre-trained VLMs (InstructBLIP (Dai et al., 2024) with Flan-T5 (Chung et al., 2024) or Vicuna (Zheng et al., 2023)) to enhance policy learning for navigation and navigational reasoning.

**Instruction Synthesis.** Another method to improve the agent's generalization ability is to synthesize more instructions. Early works employ the Speaker-Follower framework (Fried et al., 2018; Tan et al., 2019; Kurita & Cho, 2020; Guhur et al., 2021a) to train an offline speaker (instruction generator) using human-annotated instruction-trajectory pairs. It then generates new instructions based on sequences of panoramas along a given trajectory. However, Zhao et al. (2021) observe that these generated instructions are low-quality and show a poor performance in human wayfinding evaluation. Marky (Wang et al., 2022a; Kamath et al., 2023) addresses this limitation using a multi-modal extension of the multilingual T5 model (Xue et al., 2020) with text-aligned visual landmark correspondences, achieving near-human quality on R2R-style paths in unseen environments. PASTS (Wang et al., 2023c) introduces a progress-aware spatial-temporal Transformer speaker to better leverage the sequenced multiple vision and action features. SAS (Gopinathan et al., 2024) generates instructions with rich spatial information using semantic and structural cues from the environment. SRDF (Wang et al., 2024c) builds a strong instruction generator with iterative self-training. Additionally, instead of training an offline instruction generator, some recent research (Liang et al., 2022; Lin et al., 2023b; Zhang & Kordjamshidi, 2023; Wang et al., 2023e; Magassouba et al., 2021) generates instructions while navigating. For instance, LANA (Wang et al., 2023e) introduces a language-capable navigation agent that not only executes navigation instructions but also provides route descriptions.

# 5 VLN Agent: Learning an Embodied Agent for Reasoning and Planning

While the world and human models empower visual and language understanding abilities, VLN agents need to develop embodied reasoning and planning capabilities to support their decision-making. From this

perspective, we discuss two challenges: grounding and reasoning, and planning. We also explore the method of directly applying foundation models as the VLN agent backbone.

## 5.1 Grounding and Reasoning

Different from other VL tasks, such as VQA and Image Captioning, which primarily focus on static alignment between images and corresponding textual descriptions, the VLN agent needs to reason about spatial and temporal dynamics in the instructions and the environment based on its actions. Specifically, the agent should consider previous actions, identify the part of the sub-instruction to execute, and ground the text to the visual environment to execute the action accordingly. Previous methods primarily rely on explicit semantic modeling or auxiliary task design to obtain such abilities. However, pre-training with specially designed tasks has become the dominant approach with the advent of foundation models.

**Explicit Semantic Grounding.** The previous efforts enhance the agent's grounding ability through explicit semantic modeling in both vision and language modalities, including modeling motions and landmarks (Hong et al., 2020b; He et al., 2021; Hong et al., 2020a; Zhang et al., 2021b; Qi et al., 2020a), utilizing syntactic information in the instruction (Li et al., 2021), as well as spatial relations (Zhang & Kordjamshidi, 2022b; An et al., 2021). Very few works (Lin et al., 2023a; Zhan et al., 2024a; Wang et al., 2023b) explore explicit grounding in the VLN agent with the foundation models. Lin et al. (2023a) proposes actional atomic-concept learning and map visual observations to faciliate multi-modal alignments.

**Pre-training VLN Foundation Models.** Except for explicit semantic modeling, the previous research also enhances the agent's grounding ability through auxiliary reasoning tasks (Ma et al., 2019; Wu et al., 2021; Zhu et al., 2020; Raychaudhuri et al., 2021; Dou & Peng, 2022; Kim et al., 2021). Such methods are less explored in VLN agents with foundation models, as their pre-training already provides a general understanding of spatial and temporal semantics prior to navigation. Various pre-training methods with specially designed tasks have been proposed to improve the agent's grounding ability. Lin et al. (2021) introduce pre-training tasks specifically designed for scene and object grounding. LOViS (Zhang & Kordjamshidi, 2022a) formulates two specialized pre-training tasks to enhance orientation and visual information separately. HOP (Qiao et al., 2022; 2023a) introduces a history-and-order aware pre-training paradigm that emphasizes historical information and trajectory orders. Li & Bansal (2023) suggests that enhancing the agent with the ability to predict future view semantics helps the agent in longer path navigation performance. Dou et al. (2023) design a masked path modeling objective to reconstruct the original path given a randomly masked sub-path. Cui et al. (2023) propose entity-aware pre-training by predicting grounded entities and aligning them to text.

## 5.2 Planning

Dynamic planning enables VLN agents to adapt to environmental changes and improve navigation strategies on the fly. Alongside the graph-based planners that utilize global graph information to enhance local action spaces, the rise of foundational models, particularly LLMs, has brought LLM-based planners into the VLN field. These planners use LLMs' vast commonsense knowledge and advanced reasoning to create dynamic plans that improve decision-making.

**Graph-based Planner.** Recent advancements in VLN emphasize enhancing navigational agents' planning capabilities through global graph information. Among them, Wang et al. (2021); Chen et al. (2022c); Deng et al. (2020); Zheng et al. (2024b) enhance the local navigation action spaces with global action steps from graph frontiers of visited nodes for better global planning. Gao et al. (2023) further enhances navigation decision-making with high-level planning for zone selection and low-level planning for node selection. Moreover, Liu et al. (2023a) enriches the graph-frontier-based global and local action spaces with grid-level actions for more accurate action prediction. In continuous environments, Krantz et al. (2021); Hong et al. (2022); Anderson et al. (2021) adopt a hierarchical planning approach utilizing high-level action spaces instead of low-level ones by selecting a local waypoint from a predicted local navigability graph. CM2 (Georgakis et al., 2022) facilitates trajectory planning by grounding instructions within a local map. Expanding on this

strategy, An et al. (2024; 2023); Wang et al. (2023g); Chang et al. (2024); Wang et al. (2022c) construct a global topological graph or grid maps to facilitate map-based global planning. Additionally, Wang et al. (2023a; 2024a) predict multiple future waypoints using either a video prediction model or a neural radiance representation model to plan the best action based on the long-term effects of predicted candidate waypoints.

**LLM-based Planner.** In parallel, some studies leverage common-sense knowledge from LLMs to generate text-based plans (Huang et al., 2022; 2023b). LLM-Planner (Song et al., 2023) creates detailed plans composed of sub-goals, dynamically adjusting these plans in real-time by integrating detected objects according to predefined program patterns. Similarly, Mic (Qiao et al., 2023b) and $A^2$Nav (Chen et al., 2023b) specialize in breaking down navigation tasks into detailed textual instructions, with Mic generating step-by-step plans from both static and dynamic perspectives, while $A^2$Nav uses GPT-3 to parse instructions into actionable sub-tasks. ThinkBot (Lu et al., 2023) employs thought chain reasoning to generate missing actions with interactive objects. VL-Map (Huang et al., 2023a) decomposes navigation instructions into sequential, goal-related functions in code format using code-written LLMs (following the Code-as-Policy (Liang et al., 2023) framework) and utilizes a dynamically built, queryable map to guide the execution of these goals. Additionally, SayNav (Rajvanshi et al., 2024) builds a 3D scene graph of the explored environment as input to LLMs for generating feasible and contextually appropriate high-level plans for the navigator.

## 5.3 Foundation Models as VLN Agents

The architecture of VLN agents has undergone significant transformations with the advent of foundation models. Initially conceptualized by Anderson et al. (2018), VLN agents were formulated within a Seq2Seq framework, employing an LSTM and an attention mechanism to model the interaction between vision and language modalities. With the advent of foundation models, the agent backend has transitioned from LSTM to Transformer and, more recently, to these large-scale pre-trained systems.

**VLMs as Agents.** The mainstream methodology leverages single-stream VLMs as the core structure of VLN agents (Hong et al., 2021; Qi et al., 2021; Moudgil et al., 2021; Zhao et al., 2022). These models process inputs from language, vision, and historical tokens simultaneously at each time step. It performs self-attention over these cross-modal tokens to capture the textual-visual correspondence, which is then used to infer the action probability. In the zero-shot VLN, CLIP-NAV (Dorbala et al., 2022) utilizes CLIP to obtain natural language referring expressions that describe the target object and make sequential navigational decisions. VLN-CE agents (Krantz et al., 2020) differentiate themselves from the VLN-DE (Anderson et al., 2018) agents by their action space, executing low-level controls in the continuous environment instead of graph-based high-level actions of view selection. Despite early works (Krantz et al., 2020; Raychaudhuri et al., 2021) utilizing LSTM to infer low-level actions, the introduction of waypoint predictors has allowed to transfer methods from DE to CE (Krantz et al., 2021; Krantz & Lee, 2022; Hong et al., 2022; Anderson et al., 2021; An et al., 2022; Zhang & Kordjamshidi, 2024). All these methods use a waypoint predictor to obtain a local navigability graph, allowing foundation models in DE to adapt to the continuous environment. In particular, the waypoint detection process primarily involves using visual observations (*e.g.*, panoramic RGBD images) to predict navigable candidate adjacent waypoints from the agent's current position as possible targets. Given the predicted waypoints, the agent selects one as the current destination.

**LLMs as Agents.** Since LLMs have powerful reasoning ability and semantic abstraction of the world, and also show strong generalization ability in unknown large-scale environments, recent research in VLN has started to directly employ LLMs as agents to complete navigation. Typically, visual observations are converted into textual descriptions and fed into the LLM along with instructions, which then perform action predictions. Innovations such as NavGPT (Zhou et al., 2024a) and MapGPT (Chen et al., 2024a) demonstrate the feasibility of zero-shot navigation, with NavGPT autonomously generating actions using GPT-4 and MapGPT converting topological maps into global exploration hints. DiscussNav (Long et al., 2024b) extends this approach by deploying multiple domain-specific VLN experts to automate and reduce human involvement in navigation tasks. It includes Instruction Analysis Experts, Vision Perception Experts, Completion Estimation Experts, and Decision Testing Experts. The use of multiple domain-specific VLN experts distributes tasks among specialized agents, reducing the burden on a single model and allowing

for optimized, task-specific processing. This multi-expert approach enhances robustness, transparency, and overall performance by leveraging the collective strengths of multiple large models. MC-GPT (Zhan et al., 2024b) employs memory topology maps and human navigation examples to diversify strategies, while InstructNav (Long et al., 2024a) breaks navigation into sub-tasks with multi-sourced value maps for effective execution. In contrast to zero-shot usage, some works (Zheng et al., 2024a; Zhang et al., 2024a; Pan et al., 2024) fine-tune LLMs to address the embodied navigation tasks effectively. Some studies have incorporated the Chain-of-Thought (CoT) (Wei et al., 2022) reasoning mechanism to improve the reasoning process. Nav-CoT (Lin et al., 2024a) transforms LLMs into a world model and navigational reasoning agent, streamlining decisions by simulating future environments. This demonstrates the flexibility and practical potential of fine-tuned language models in both simulation and real-world scenarios, marking a significant advancement over traditional applications.

## 6 Challenges and Future Directions

While foundation models have enabled novel solutions to VLN, several limitations remain under-explored, and new challenges arise. In this section, we outline the challenges and future direction of VLN from the perspectives of benchmarks, the world model, the human model, the agent model, and real robot deployment.

**Benchmarks: Limitations of Data and Task.** The current VLN datasets have limitations regarding quality, diversity, bias, and scalability. For example, in the R2R dataset, the instruction-trajectory pairs are biased to the shortest path, which may not accurately represent real-world navigation scenarios. We discuss the trends and recommendations on how VLN benchmarks can be improved.

- *Unified and Realistic Tasks and Platforms.* Establishing robust benchmarks and ensuring reproducibility are crucial for evaluating VLN in real-world settings. Real-world variability necessitates comprehensive benchmarks reflecting navigation challenges. A universal sim-to-real evaluation platform, like OVMM (Yenamandra et al., 2023), is needed for standardized testing across simulated and real-world settings. In addition, the tasks and activities should be realistic and designed originated from human needs. For instance, BEHAVIOR-1K (Li et al., 2024a) presents a benchmark of everyday household activities in virtual, interactive, and ecological environment to address the demands for diversity and realism.

- *Dynamic Environment.* Real-world environments are inherently complex and dynamic, with moving objects, people, and variations like lighting and weather presenting unexpected situations (Ma et al., 2022). These factors disrupt the visual perception of navigation systems and make maintaining reliable performance difficult. Recent efforts like HAZARD (Zhou et al., 2024c), Habitat 3.0 (Puig et al., 2024), and HA-VLN (Li et al., 2024b) consider dynamic environments and provide a good starting point.

- *Indoors to Outdoors.* VLN agents navigating in outdoor environments, *e.g.*, autonomous driving and aerial vehicles, also start to get more attention (Vasudevan et al., 2021; Li et al., 2024c), with various language-guided datasets (Sriram et al., 2019; Ma et al., 2022) developed. Early studies have attempted to involve LLMs in these tasks, either with prompt engineering (Shah et al., 2023; Sha et al., 2023; Wen et al., 2023), or by fine-tuning LLMs to predict the next action or plan future trajectories (Chen et al., 2024b; Mao et al., 2023). To adapt off-the-shelf VLMs to these outdoor navigation domains, real-world driving videos (Xu et al., 2024a; Yuan et al., 2024), simulated driving data (Wang et al., 2023d; Shao et al., 2024) and them both (Sima et al., 2023; Huang et al., 2024b) have been utilized for instruction tuning so that these foundation models learn to predict future throttle and steering angles. Additional reasoning and planning modules have also been integrated into foundation model driving agents (Huang et al., 2024b; Tian et al., 2024). We refer the readers to surveys and position papers for a detailed review (Li et al., 2023b; Cui et al., 2024; Gao et al., 2024; Yan et al., 2024).

**World Model: From 2D to 3D.** Building effective world representations is a central research theme in embodied perception, reasoning, and planning. VLN is fundamentally a 3D task, where the agent perceives the real-world environment in 3D. Although the current research represents the world with strong and generic 2D representations, they fall short of spatial language understanding in the 3D world (Zhang et al., 2024d).

Many explicit 3D representations are developed in prior work, including various semantic SLAMs and volumetric representation (Chaplot et al., 2020; Min et al., 2021; Saha et al., 2022; Blukis et al., 2022; Zhang et al., 2022b; Liu et al., 2024), depth information (An et al., 2023), Bird's-Eye-View representations like grid map (Wang et al., 2023g; Liu et al., 2023a), and local metrics map (An et al., 2023). These representations are limited because they reduce the object set to a closed set, making them inadequate for open-vocabulary settings with natural language. Several studies develop queryable map/scene representations by integrating multi-view image features captured from CLIP into 3D voxel grids (Jatavallabhula et al., 2023; Chang et al., 2023) or top-down feature maps (Huang et al., 2023a; Chen et al., 2023a), as well as utilizing scene graphs (Rana et al., 2023; Gu et al., 2024) to represent spatial relationships. However, adapting 3D representations learned from large-scale data for VLN agents to better perceive the 3D environment is still under exploration. The recent rise of 3D foundation models (Hong et al., 2023b; Huang et al., 2024a; Chen et al., 2024d;e), including 3D reconstruction models (Hong et al., 2024) and 3D multimodal representations (Yang et al., 2024; Zhang et al., 2024c;e), can be crucial for VLN.

**Human Model: From Instruction to Dialogue.** Previous efforts predominantly adopt either a speaker-listener paradigm or restricted QA dialogue (Thomason et al., 2020; Gao et al., 2022) that only allows the agent to ask for help. Recently, there has been a surge in new benchmarks featuring open-ended dialogue instructions (De Vries et al., 2018; Banerjee et al., 2021; Padmakumar et al., 2022; Ma et al., 2022; Fan et al., 2023a), supporting fully free-form communication where agents can ask, propose, explain, suggest, clarify, and negotiate even in ambiguous or confusing scenarios. Still, current approaches rely on rule-based dialogue templates to tackle these complexities (Zhang et al., 2023; Parekh et al., 2023; Gu et al., 2023), though they might feature a foundation model component. Huang et al. (2024b) perform conversational tuning on a video-language model using human-human dialogue data paired with simulated navigation videos, showcasing enhanced dialogue generation capabilities while navigation. Moving forward, it is imperative for future research to integrate foundation models for situated task-oriented dialogue management (Ulmer et al., 2024), or explore existing foundation models for task-oriented dialogue (He et al., 2022).

**Agent Model: Adapting Foundation Models for VLN.** While foundation models show strong generalizability, incorporating them into navigation tasks remains challenging. LLMs fundamentally lack the capability to visually perceive the actual environment and are prone to hallucinations. We als discuss capabilities of LLMs in planning and reasoning.

- *Lack of Embodied Experience.* This limitation can lead to scenarios where LLMs rely solely on pre-established commonsense for task planning and reasoning, which might not meet specific real-world needs (Xiang et al., 2024). Some pipelines tackle this issue by captioning the visual observations to textual descriptions as prompts for LLMs (Zheng et al., 2022), with a potential loss of essential visual semantics. Compared with LLMs, VLM agents demonstrate the potential to perceive the visual world and plan (Zhang et al., 2024a). Still, these models are primarily developed from internet data, which lack embodied experiences (Mu et al., 2024) and need finetuning for robust agentic decision-making (Zhai et al., 2024). Further research is needed to transfer the commonsense knowledge in foundation model agents to generalize to embodied situations. Recently proposed embodied foundation models (such as EmbodieGPT (Mu et al., 2024), PaLM-E (Driess et al., 2023) and Octopus (Yang et al., 2025)) offer a promising solution for enabling agents to operate more effectively in interactive environments. They fine-tune foundation models across multiple embodied tasks to bridge the gap between an agent's understanding of vision, language, and embodied actions, enhancing the foundation model's ability to comprehend and execute based on multimodal input.

- *Hallucination Issue.* LLMs and VLMs might generate non-existent objects, leading to misinformation (Li et al., 2023c; Chen et al., 2024c). For example, when LLM performs task planning, it may generate instructions such as "*go forward and turn left at the sofa*" even if there is no sofa in the room. This inaccuracy may cause them to execute incorrect or impossible actions.

- *LLMs in Planning and Reasoning.* There are some literatures evaluating the zero-shot reasoning and planning capabilities of LLMs, particularly in relation to PlanBench (Valmeekam et al., 2022) and CogEval (Momennejad et al., 2023), which highlight LLMs' limitations in more complex planning tasks. These works assess LLMs in a variety of challenging settings, such as plan generation, optimality, robust-

ness, and reasoning, and identify that LLMs sometimes struggle with hallucinations or fail to grasp the relational structures underlying complex planning problems. In the context of VLN, the action space and the planning requirements are relatively constrained due to the fixed indoor environments and the limited set of navigational actions. This bounded setting makes it more feasible for LLMs to provide step-by-step instructions for coarse-grained directions, which has been demonstrated to be effective in previous works. In VLN tasks, the LLM's role is not to take over the entire planning process, but rather to assist by offering a structured breakdown of instructions. The agent's actual decision-making remains primarily reliant on other components, such as perception and motion control. Therefore, in VLN tasks, the LLM's planning serves more as a supplementary guide rather than the sole decision-making factor.

**Deployment: From Simulation to Real Robots.** Simulated settings often lack the complexity and variability of real-world environments, and lower-quality rendered images exacerbate this issue. First, the perception gap results in decreased performance and accuracy, highlighting the need for more robust perception systems. Wang et al. (2024b) have started to explore the use of semantic maps and 3D feature fields to provide monocular robots with panoramic perception shows improved performance. The embodiment gap and the data scarcity are also bottlenecks. The rise of robot teleportation (He et al., 2024b) also provides an alternative to scale up VLN data for foundation models in real human-robot communications.

## 7 Broader Impact

Foundation models hold great promise for advancing vision-language navigation. However, it is essential to address their broader ethical, legal, and societal implications. Given that they are pre-trained on vast, web-scale datasets, these models can carry inherent biases, which may result in fairness concerns, *e.g.*, to multilingual users. Some approaches involve continual model training, it is critical to acknowledge and mitigate any potential risks to user privacy, especially when deployed in real-world applications such as home robotics.

## 8 Acknowledgement

This work is supported in part by the ARO Award W911NF2110220, NSF grant IIS-1949634, and ONR grant N00014-23-1-2417 & N00014-23-1-2356. Any opinions, findings, and conclusions or recommendations expressed in this material are those of the authors and do not necessarily reflect the views of the funding agencies.

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
