# OpenReview forum: "Vision-and-Language Navigation Today and Tomorrow: A Survey in the Era of Foundation Models"
_TMLR — Accepted by TMLR_

### Review · Reviewer_HbEf · 2024-08-12

**Summary Of Contributions:**

The paper represents a survey of the role of foundation models in vision-and-language navigation (VLN). It analyses the topic based on the 3 areas of VLN (world model, human model, and VLN agent). It also includes an overview of existing benchmarks. Besides discussing multiple existing methods and how they differ in terms of the usage of foundation models it also includes some perspective of open challenges and potential for future research.

**Audience:**

Yes

**Broader Impact Concerns:**

Does not have a broader impact statement. It could a general one be added as foundation models are very powerful, but can also be biased.

**Claims And Evidence:**

Yes

**Requested Changes:**

- The paper would benefit from partially improved representation of the content, especially when currently talking only generally about foundation models. Given that this is the main topic, it should be clear what kind of foundation model (e.g., which modalities) is used. Especially Section 3 is lacking in this regard.
- Instead of giving a tabular overview of the datasets, something like this focusing on the main content of the paper, i.e., section 3-5, and representing the content in a structured way would be more helpful for the reader.

**Strengths And Weaknesses:**

**Strengths**
- Paper mostly well structured (see also weaknesses) and written.
- Covers many different approaches.
- Given the rise of foundation models and there increased importance and success, a survey paper is very valuable.
- The section about challenges and future directions is interesting and can inspire future research.

**Weaknesses**
- The paper often refers to general foundation models, without specifying which kind of. For example last paragraph of Section 3. Giving a more strucutured overview and/or being more specific of which foundation models, covering which modalities is used for which task would be helpful, e.g., as done in section 5.3, or even providing a tabular overview if possible. However, this structure, focusing on the foundation model is not consistent througout the paper. E.g., in 3.1. it is less clear. There it seems to be more general about transformer based architecture and not specifically foundation models. This separation should be clearer.
- The paragraph and table 1 about benchmarks is a bit disconnected from the rest of the story line. That part is less connected to foundation models. The same applies to the part about evaluation metrics. Also there the connection to foundation model and the purpose of the paragraph for the whole paper is unclear.

---

> ### Author Response · Authors · 2024-10-09
> **Response to HbEf**
>
> We appreciate the reviewer's recognition of the strengths of our work, including our clear structure, comprehensive coverage of various approaches, the value of our survey, and the meaningful discussion of future directions. We address the reviewer's comments as follows and please review our revised paper.
>
> ### **Section2: Background**
> ### 1. Instead of giving a tabular overview of the datasets, something like this focusing on the main content of the paper.
>
> We appreciate the reviewer pointing this out. We address this issue by clarifying the benchmarks in **Section 2.3** to better connect the dataset and LAW framework. We mainly clarify from the following two perspectives: (1) Unlike other multimodal tasks such as VQA, which have a relatively fixed task definition and format, VLN encompasses a wide range of benchmarks and task formulations. These variations include differences in environments, action spaces, and language formats. These distinctions introduce unique challenges in addressing the broader VLN task and must be clearly understood as prerequisites for developing effective methods using different foundation models. (2) VLN benchmarks are categorized based on several key aspects outlined in the LAW framework, where foundation models serve as the backbone for both world modeling and agent behavior. This framework provides a comprehensive view of reasoning and planning within foundation models, closely aligning with the core challenges faced in VLN.
>
> ### **Section3: World Model**
> ### 1.  The paper would benefit from partially improved representation of the content. What kind of foundation model is used. Especially Section 3 is lacking in this regard.
>
> We have rewritten History Encoding in **Section 3.1**. We mention in general how foundation models are used in VLN encoding and decision-making and then discuss the architecture variants used for history encoding and graph-based history encoding.
>
>
> ### **Broader Impact**
> We have included a Broader Impact section in our paper, which discusses the potential societal, ethical, and practical implications of our work.

---

### Review · Reviewer_Eo4S · 2024-08-15

**Summary Of Contributions:**

This paper presents a comprehensive summary of vision and language navigation research, with a special focus on the role of pretrained foundation models. The paper presents a useful taxonomy for organizing the literature, dividing the problem into three subproblems involving the formation of world models, human models, and the implementation of the navigation agent itself. A useful taxonomy is also presented for characterizing the range of datasets and tasks that have been considered in this literature. Finally, the paper identifies several limitations of current work and avenues for future work.

**Audience:**

Yes

**Broader Impact Concerns:**

I did not identify any ethical implications of this work that would necessitate a Broader Impact Statement.

**Claims And Evidence:**

Yes

**Requested Changes:**

It would be helpful to discuss the implications of findings indicating poor zero-planning in LLMs for their potential use as VLN agents.

**Strengths And Weaknesses:**

I found this review to be very comprehensive, and appreciated the clear organizational framework. I think it will serve as a very useful survey of this field and the ways in which it is being influenced by pretrained foundation models. I have very little to say in terms of weaknesses, but do have just a couple of comments below that may be helpful for further improving the paper.

Comments:
- My main comment is that it is stated several times in the paper that LLMs provide powerful reasoning and planning capabilities that may aid VLN agents. I can see how this may be a reasonable characterization relative to the previous state-of-the-art in this field (i.e. prior to the adoption of pretrained foundation models), but there is not much discussion of the extensive literature and controversy surrounding the zero-shot reasoning and planning capabilities of these types of systems. Of direct relevance to the VLN domain is recent work showing that LLMs have poor zero-shot planning abilities [1,2]. It would be helpful to discuss the implications of these findings for the use of such systems as VLN agents.
- One very minor comment is that I found it odd to use the word 'milestone' as a verb, as this isn't a usage that I have seen before. It might be more straightforward to say something like 'document the progress' rather than 'milestone the progress', but again this is a minor comment.

[1] Valmeekam, K., Marquez, M., Olmo, A., Sreedharan, S., & Kambhampati, S. (2024). Planbench: An extensible benchmark for evaluating large language models on planning and reasoning about change. Advances in Neural Information Processing Systems, 36.

[2] Momennejad, I., Hasanbeig, H., Vieira Frujeri, F., Sharma, H., Jojic, N., Palangi, H., ... & Larson, J. (2024). Evaluating cognitive maps and planning in large language models with CogEval. Advances in Neural Information Processing Systems, 36.

---

> ### Author Response · Authors · 2024-10-09
> **Response to Eo4S**
>
> We sincerely appreciate the reviewer’s acknowledgment of the strength of our review, particularly highlighting the clear organizational framework and its value to the community. We have addressed the reviewer's comments as follows and please review our revised paper.
>
> ### **Section6: Challenges and Future Direction**
> ### 1. More discussion about zero-shot LLM .
>
> Thank you for your valuable feedback. There are some literature evaluating the zero-shot reasoning and planning capabilities of LLMs, particularly in relation to PlanBench (Valmeekam et al., 2022) and CogEval (Momennejad et al., 2023), which highlight LLMs’ limitations in more complex planning tasks. These works assess LLMs in a variety of challenging settings, such as plan generation, optimality, robustness, and reasoning, and identify that LLMs sometimes struggle with hallucinations or fail to grasp the relational structures underlying complex planning problems. In the context of VLN, the action space and the planning requirements are relatively constrained due to the fixed indoor environments and the limited set of navigational actions. This bounded setting makes it more feasible for LLMs to provide step-by-step instructions for coarse-grained directions, which has been demonstrated to be effective in previous works. In VLN tasks, the LLM’s role is not to take over the entire planning process, but rather to assist by offering a structured breakdown of instructions. The agent’s actual decision-making remains primarily reliant on other components, such as perception and motion control. Therefore, in VLN tasks, the LLM’s planning serves more as a supplementary guide rather than the sole decision-making factor. We have added this discussion in **Section 6**.
>
>
> ### 2.  Document the progress instead of milestone the progress.
>
> We appreciate the reviewer pointing this out. We have revised the ``milestone`` to the``document`` the progress in the **Abstract**.

---

> > ### Comment · Reviewer_Eo4S · 2024-10-10
> >
> > Thanks to the authors for these replies. I continue to recommend acceptance for the paper.

---

### Review · Reviewer_s3ya · 2024-09-27

**Summary Of Contributions:**

1. This paper surveys the field of vision-language navigation (VLN) with an emphasis on foundation models.
2. It provides a brief summary of the VLN setup, the benchmarks in the field, and their taxonomy.
3. It leverages the LAW framework to outline the three components of a VLN agent (world model, human model, VLN agent). Subsequently, it summarises the literature addressing each of these components, including both the traditional approaches as well as those involving foundation models.
4. The paper identifies challenges and directions for future work in this area.

**Audience:**

Yes

**Claims And Evidence:**

Yes

**Requested Changes:**

These changes could strengthen the work:

1. Addressing the weaknesses.
2. The evaluation metrics could be explained better. For example, CLS has not been explained at all. Does SPL aim to provide higher scores for longer trajectories?
3. Section 3 (world model): LLMs have strong commonsense and world knowledge. Have existing works leveraged this to build better world models? I’m curious to hear the authors’ thoughts on this.
4. Section 4.1 (information seeking): Most of the papers discussed here involve using LLMs as an oracle. Are there works that leverage LLMs for deciding when to ask for help and for asking informative questions?
5. Section 4.2 (Pre-trained representations): Most of the works in this section are from 2020 or before. Are there more recent works in this direction?
6. Section 3.2 (Environmental Augmentation): In the last paragraph, the authors say that the paradigm has shifted from using augmented environments in the fine-tuning stage to the pre-training stage. While in general I understand the different between the two stages, it’s not clear to me what it means in this context? Is it that the scale of data being used now (in the pre-training stage) is much larger than before (in the fine-tuning stage)? or is there more to it. Some clarification here would be helpful.
7. Section 3.1 (History Encoding): The references involving panorama encoders could be re-written to explain how panorama encoders work and are useful for VLN.
8. Section 4.1 (Information Seeking): the references for using foundation models as oracle models (Fan et al., Lin et al., Ye et al., Zhu et al.) could be re-written to enhance clarity.
9. Section 5.3 (VLMs as agents): The content regarding waypoint detection was not clear. Perhaps a brief explanation of waypoint detection would be helpful for the reader.
10. Section 5.3 (LLMs as agents): DiscussNav – why is a multi-agent approach useful?
11. Some other references that could be useful: [1], [2] (benchmarks), [3]
12. [typo] Section 3.1: “Prior to employing the foundation models in VLN, LSTM hidden states serve…”, grammatically, the sentence reads a bit off (using present tense following the word ‘prior’). It’s possible that the grammar is technically okay, just wanted to flag it in case this was unintentional.
13. [typo] Section 4.1: “Zhang & Kordjamshidi (2023)..” (two periods)


=== References ===

[1] Sarch, Gabriel, et al. "Open-Ended Instructable Embodied Agents with Memory-Augmented Large Language Models." Findings of the Association for Computational Linguistics: EMNLP 2023. 2023.

[2] Li, Chengshu, et al. "Behavior-1k: A benchmark for embodied ai with 1,000 everyday activities and realistic simulation." Conference on Robot Learning. PMLR, 2023.

[3] Liang, Jacky, et al. "Code as policies: Language model programs for embodied control." 2023 IEEE International Conference on Robotics and Automation (ICRA). IEEE, 2023.

[4] Yang, Jingkang, et al. "Octopus: Embodied vision-language programmer from environmental feedback." arXiv preprint arXiv:2310.08588 (2023).

[5] Driess, Danny, et al. "PaLM-E: An Embodied Multimodal Language Model." International Conference on Machine Learning. PMLR, 2023.

**Strengths And Weaknesses:**

=== Strengths ===
1. The paper is well-organized. The authors take a top-down approach and divide the paper into primarily three sections (world model, human model, VLN agent), which is intuitive. Further, within each section, they identify the key role(s) of each component (e.g., the world model needs to encode the history of the actions and observations made by the agent) and describe how previous works have approached them. Wherever possible, they also discuss how foundation models can help improve performance in this respect. It neatly organizes the VLN literature, which is crucial for a survey.
2. The paper is well-written, with most parts being easy to follow.
3. The paper identifies important challenges and future directions.


=== Weaknesses ===
1. Individual subsections are a bit dense and it is hard to understand the core ideas from the papers from the survey alone (especially for people with limited background in VLN/ foundation models). I do understand that it might be hard for the authors to explain the ideas in more detail given the space limit.
2. The reader might benefit from a brief introduction to relevant foundation models, i.e., the ones that have been applied to VLN, such as CLIP, LLMs, etc. Perhaps the paper could point to some surveys on these models or provide a brief overview.
3. I didn’t see much emphasis on multimodal LLMs (like GPT-4o). Have some works tried to use such models in the VLN setting? Some examples could be [4], [5]. I would urge the authors to discuss more works using multimodal LLMs if applicable.


=== References ===

[1] Sarch, Gabriel, et al. "Open-Ended Instructable Embodied Agents with Memory-Augmented Large Language Models." Findings of the Association for Computational Linguistics: EMNLP 2023. 2023.

[2] Li, Chengshu, et al. "Behavior-1k: A benchmark for embodied ai with 1,000 everyday activities and realistic simulation." Conference on Robot Learning. PMLR, 2023.

[3] Liang, Jacky, et al. "Code as policies: Language model programs for embodied control." 2023 IEEE International Conference on Robotics and Automation (ICRA). IEEE, 2023.

[4] Yang, Jingkang, et al. "Octopus: Embodied vision-language programmer from environmental feedback." arXiv preprint arXiv:2310.08588 (2023).

[5] Driess, Danny, et al. "PaLM-E: An Embodied Multimodal Language Model." International Conference on Machine Learning. PMLR, 2023.

---

> ### Author Response · Authors · 2024-10-09
> **Response to Reviewer s3ya**
>
> We appreciate the reviewer's recognition of our strengths and the constructive feedback. We have addressed the comments as follows. Please review our revised paper.
>
> ### **Individual subsections are a bit dense and it is hard to follow by people with limited background in VLN.**
>
> Thank you for pointing out that certain sections of the paper may be challenging for readers with a limited background in VLN. We understand the importance of making our work accessible to a broader audience. To improve readability, we have revised the paper based on the reviewer’s detailed feedback to clarify key concepts, such as foundation models and VLN metrics in Section 2, history encoding in Section 3, information seeking in Section 4, and the waypoint predictor in Section 5. We hope these revisions enhance the clarity and readability of the paper.
>
>
>
>
> ### **Section2: Background**
>
> ### 1. Introduction to relevant foundation models.
>
> We have included a brief introduction to foundation models in **Section 2.4** to introduce key foundation models and point to the relevant surveys.
>
> ### 2. CLS has not been explained at all. Does SPL aim to provide higher scores for longer trajectories?
> We have revised **Section2.3** to provide explanations of CLS and SPL. SPL is designed to reward efficient trajectories that are as close as possible to the shortest path to the target. If the agent takes a longer trajectory than necessary, the SPL score is penalized.
>
>
> ### **Section3: World Model**
>
>
> ### 1. Have existing works leveraged this to build better world models?
>
> We have modified **Section 3.1** to discuss the use of LLMs for representing history and maps.  Essentially, the environments are converted into text descriptions. The navigation history is encoded as a sequence of image descriptions, together with relative heading/elevation/distance information (NavGPT), or further enhanced with topological graph information (MapGPT).
>
> ### 2. Clarification of pre-training and fine-tuning stages.
>
> We have modified the last paragraph in **Section 3.2**.  There are two key differences. First, before the rise of pre-training in foundation models, navigation agents learned directly from instruction-trajectory data using imitation learning and reinforcement learning, without a pre-training stage. Second, as the reviewer correctly noted, the scale of data used now is significantly larger. The augmented environments have expanded from 61 room environments to over 800, and the data scale has grown from 7K (Liu et al., 2021) examples to 4.9M (Wang et al., 2023).
>
> ### 3. re-written panorama encoders.
>
> We have modified history encoding in **Section3.1** to explain how panoramic encoder works.
>
>
> ### **Section4: Human Model**
>
> ### 1.  LLMs for deciding when to ask for help and for asking informative questions.
>
> We have provided works discussing using LLMs for informative questions in **Section 4.1**,  such as CP, ICL and VLN-Copilot.
>
>
> ### 2. Re-written references for using foundation models as oracle models.
>
> Please see our rewritten version in Information Seeking of **Section 4.1**.
>
> ### 3.  Most of the works in this section are from 2020 or before. Are there more recent works in this direction?
>
> We have discussed extra works in **Section 4.2**, such as Airbert, CLEAR, ProbES, and NavGPT2.
>
>
> ### **Section5: VLN Agent**
>
> ### 1.  Clarification of Waypoint predictor.
>
> We have provided the explanation of the waypoint predictor in  **Section 5.3**.
>
>
> ### 2.  DiscussNav – why is a multi-agent approach useful?
>
> We added more details about the DiscussNav and the benefits of multiple agents in **Section 5.3**. DiscussNav effectively reduces the burden on a single decision-making model by distributing tasks among specialized experts. Moreover, the collective expertise of multiple large models enhances robustness and minimizes the risk of single-point failure, ensuring more reliable outcomes.
>
>
> ### **Section6: Challenges and Future Direction**
> ### 1. Emphasis on multimodal like gpt4o.
>
> As we discussed in **Section 6 (Agent Model)**,  there is a key limitation for LLM/VLM-based agents: multimodal LLMs often lack embodied experience. EmbodiedGPT addresses this limitation by designing specific policy networks and fine-tuning on embodied datasets. We believe PaLM-E and Octopus, as mentioned by the reviewer, fall into the same category as the EmbodiedGPT—embodied foundation model. These models present a promising solution to the above-mentioned limitation by incorporating embodied actions into foundation model training, thus bridging the gap between understanding and action execution. We have included the above discussion in the Agent Model of **Section 6**.
>
>
>
> ### **Other References and Typos**
> We have incorporated the suggested references and corrected the typos. Specifically, we added HELPER in Section 3.1, VL-Map in Section 5.2, and BEHAVIOR-1K in Section 6.

---

### Decision · Action_Editor_jUTM · 2024-11-15

**Recommendation:** Accept as is

**Comment:**

The paper is fine as is, but for the camera-ready, remember to remove the blue-colored text which identifies the changes (useful for the revision, but should not be camera-ready).

**Audience:**

Yes, all three reviewers agree on that.

VLN navigation is relevant for many researchers in the field.

**Claims And Evidence:**

Yes, all three reviewers agree on that.

The paper sets out to survey the field of VLN navigation, and I agree with the reviewers that they succeeded.